# Synthesis of Tricyclic Pterolobirin H Analogue: Evaluation of Anticancer and Anti-Inflammatory Activities and Molecular Docking Investigations

**DOI:** 10.3390/molecules28176208

**Published:** 2023-08-23

**Authors:** Houda Zentar, Fatin Jannus, Marta Medina-O’Donnell, Az-eddine El Mansouri, Antonio Fernández, José Justicia, Enrique Alvarez-Manzaneda, Fernando J. Reyes-Zurita, Rachid Chahboun

**Affiliations:** 1Departamento de Química Orgánica, Facultad de Ciencias, Instituto de Biotecnología, Universidad de Granada, 18071 Granada, Spain; zentarhouda@correo.ugr.es (H.Z.); ajfvargas@ugr.es (A.F.); jjusti@ugr.es (J.J.); eamr@ugr.es (E.A.-M.); 2Departamento de Bioquímica y Biología Molecular I, Facultad de Ciencias, Universidad de Granada, 18071 Granada, Spain; fatin@correo.ugr.es (F.J.); mmodonnell@ugr.e (M.M.-O.); 3Chemistry Department, University of the Free State, P.O. Box 339, Bloemfontein 9300, South Africa; elmansouri.a@ufs.ac.za

**Keywords:** function-oriented synthesis (FOS), cassane diterpenoids, anticancer, anti-inflammatory, docking

## Abstract

Pterolobirin H (**3**), a cassane diterpene isolated from the roots of Pterolobium macropterum, exhibits important anti-inflammatory and anticancer properties. However, its relatively complex tetracyclic structure makes it difficult to obtain by chemical synthesis, thus limiting the studies of its biological activities. Therefore, we present here a short route to obtain a rational simplification of pterolobirin H (**3**) and some intermediates. The anti-inflammatory activity of these compounds was assayed in LPS-stimulated RAW 264.7 macrophages. All compounds showed potent inhibition of NO production, with percentages between 54 to 100% at sub-cytotoxic concentrations. The highest anti-inflammatory effect was shown for compounds **15** and **16**. The simplified analog **16** revealed potential NO inhibition properties, being 2.34 higher than that of natural cassane pterolobirin H (**3**). On the other hand, hydroxyphenol **15** was also demonstrated to be the strongest NO inhibitor in RAW 264.7 macrophages (IC_50 NO_ = 0.62 ± 0.21 μg/mL), with an IC_50NO_ value 28.3 times lower than that of pterolobirin H (**3**). Moreover, the anticancer potential of these compounds was evaluated in three cancer cell lines: HT29 colon cancer cells, Hep-G2 hepatoma cells, and B16-F10 murine melanoma cells. Intermediate **15** was the most active against all the selected tumor cell lines. Compound **15** revealed the highest cytotoxic effect with the lowest IC_50_ value (IC_50_ = 2.45 ± 0.29 μg/mL in HT29 cells) and displayed an important apoptotic effect through an extrinsic pathway, as evidenced in the flow cytometry analysis. Furthermore, the Hoechst staining assay showed that analog **15** triggered morphological changes, including nuclear fragmentation and chromatin condensation, in treated HT29 cells. Finally, the in silico studies demonstrated that cassane analogs exhibit promising binding affinities and docking performance with iNOS and caspase 8, which confirms the obtained experimental results.

## 1. Introduction

It is well known that natural products constitute the main axis for the rational design of new drugs [1,2]. However, the scarcity of these in their natural sources and their structural complexity is, in many cases, an important limitation to evaluating their biological potential. The preparation of simple analogs of natural products is one of the important strategies used to study the structure–activity relationship (SAR) and also to develop new drugs more economically and with lesser difficulty. Diterpenes with a cassane skeleton are an example of natural products whose biological activities have been poorly studied, largely due to their difficult chemical synthesis [3]. However, these metabolites continue to attract the attention of researchers due to the broad spectrum of their pharmacological activities, including antitumor, anti-inflammatory, antimicrobial, antiplasmodial, antimalarial, antiviral, antioxidant, antipyretic, antiperiodic, anthelmintic, antinociceptive, and antidiabetic properties [4]. In the last decade, some synthetic routes towards these types of diterpenes have been described [5,6,7,8,9]. One subclass of these diterpenes is characterized by a tetracyclic framework and by a furanic (furanoditerpenoids) or α, β-unsaturated butenolide ring fused at the C-12 and C-13 positions [4]. Representative examples of this type of compound are benthaminin 1 (**1**), which was synthesized for the first time by our group, starting from the labdane diterpene *trans*-communic acid [10], and exhibited antibacterial activity with a minimum inhibitory concentration (MIC) value of 47.8 μM for *Staphylococcus aureus* and *Micrococcus flavus* [11].

Other pioneering strategies towards aromatic cassane-type diterpenoids have been achieved by our group, such as the first synthesis of taepeenin F (**2**), starting from dehydroabietic acid and abietic acid [12,13], as well as the first enantiospecific synthesis of pterolobirin H (**3**), pterolobirin G (**4**), and (5a)-vouacapane-8(14), 9(11)-diene (**5**), starting from (+)-sclareolide (**6**) [3] (Figure 1).

These natural cassane diterpenes **3**–**5** were recently isolated from the roots of *Pterolobium macropterum* and analyzed for their anti-inflammatory potential against nitric oxide (NO) production in LPS-induced J774.A1 macrophage cells [14]. Natural furan **5** and natural pterolobirin G (**4**) did not exhibit any significant inhibition of NO production in J774.A1 cells [12]. However, Wang et al. reported that (5a)-vouacapane-8(14), 9(11)-diene (**5**) showed anti-inflammatory properties by inhibiting nitric oxide production with a ratio of 34.5% in RAW 264.7 cells at 10 μmol/L [15]. In our previous works, we have demonstrated that pterolobirin H (**3**), pterolobirin G (**4**), and (5a)-vouacapane-8(14), 9(11)-diene (**5**) showed potent inhibitory activity of NO release with IC_50NO_ values of 17.57 μg/mL, 4.01 μg /mL, and 22.02 μg /mL, respectively, in activated RAW 264.7 cells. These diterpenes also exhibited anticancer activities against HT29, Hep-G2, and B16-F10 cancer cell lines, especially pterolobirin G (**4**), with IC_50_ values of 3.87 μg /mL, 11.75 μg /mL, and 10.34 μg /mL, respectively. Pterolobirin G (**4**) displayed apoptotic effects in the HT29 cells, with a total rate of 80% of apoptosis at the IC_80_ concentration, producing a significant cell-cycle arrest in the G0/G1 phase, and with the activation of the extrinsic apoptotic pathway [16]. On the other hand, taepeenin F (**2**) also inhibited NO production in RAW 264.7 macrophages with 98% at a sub-cytotoxic concentration of 150.35 μM [13].

With such excellent biological properties, cassanes diterpenes themselves are expected to become promising leads for the development of pharmaceutical agents. However, their chemical structure is, in some cases, complex, making them difficult to obtain in quantitative terms, even by chemical synthesis, subsequently producing challenges in modifying the molecule to investigate the structure–activity relationship (SAR). To solve these synthetic drawbacks, function-oriented synthesis (FOS) [17] has been proposed as an approach to pursue the design of less complex targets with comparable or superior biological activity, obtaining potential analogs with simplified structures. In this context, some synthetic strategies for cassane diterpenoid analogs and their SAR investigations have been described; nevertheless, these analogs are still complex molecules [16,18,19]. Structurally, the cassane framework consists of five isoprene units constituting a tetracyclic structure composed of an aromatic ring linked to a five-membered heterocycle and a decalin. Taking into consideration the information above, we planned to synthesize the tricyclic analogs of natural cassanes diterpenoids. In this way, it was intended that we investigate the role that this plays in the hydrophobic part of the cassanic structure in their biological activity and, thus, advance the study of the structure–activity relationship (SAR) of these metabolites. We focused on the rational simplification of pterolobirin H (**3**) with the synthesis of a set of six simplified analogs, using a similar strategy to that used for the synthesis of pterolobirin H (**3**) from (+)-sclareolide (**6**) [3]. Only six steps are required to complete this protecting-group-free (PGF) synthesis, which relies on an unprecedented decarboxylative dienone–phenol rearrangement and a solvent-free Diels–Alder cycloaddition of diene **7** (Figure 2). In this study, we wanted to preserve the advantages of the original strategy while adding a new advantage, which is the simplification of the cassanic structure of pterolobirin H (**3**) (Figure 2A).

β-cyclocitral (**8**) is the most suitable raw material to carry out this approach, firstly due to its commercial availability at a low price; secondly, it is easily transformed into monocyclic diene **9**, an analog of diene **7**, by the Wittig reaction—the key to preparing the synthetic pterolobirin H analog **16** (Figure 2B).

Subsequently, the simplified analog and its intermediates were evaluated for their NO inhibitory activity against the lipopolysaccharide (LPS)-induced murine macrophage RAW 264.7 cell line as well as for their antiproliferative activity against HT29 (colon cancer cells), B16-F10 (melanoma cells), and HepG2 (hepatoma cells). Furthermore, SARs investigations were performed for pterolobirin H (**3**), its simplified analog, and the most active intermediate, which was then selected for additional cytometric and microscopic assays and docking studies to conduct a thorough biological investigation.

## 2. Results and Discussion

### 2.1. Chemistry

The first step of this synthetic route consists of the preparation of diene **9** from β-cyclocitral (**8**) via the Wittig reaction that was carried out at room temperature, using methyltriphenylphosphonium bromide and *n*-BuLi as the base in THF. Diene **9** was obtained in good yield (91%) [20,21]. In the next step, the Diels–Alder cycloaddition reaction of monocyclic diene **9** with dimethyl acetylenedicarboxylate (DMAD) under different conditions was studied to compare its results with those obtained with the bicyclic diene [3]. Firstly, the reaction of the cycloaddition of monocyclic diene **9** with DMAD was carried out in toluene at reflux to afford the phthalate derivative **10** and the cycloadduct **11** as a 1:3 mixture after 15 h. When the reaction was performed in xylene under reflux, the phthalate derivative **10** was obtained in good yield after 8 h. It seems clear that the increase in temperature favors the obtaining of phthalate derivative 10. In this regard, cycloadduct **11** is quantitatively transformed into **10** by heating it at 200 °C [22]. Cycloadduct **11** was obtained effectively when the reaction of cycloaddition was carried out in a sealed tube at 120 °C for 12 h. These results were similar to those described in previous studies [21,23,24]. Next, cycloadduct **11** was oxidized to dienone **12** with catalytic PDC in the presence of *tert*-butyl hydroperoxide (TBHP), in benzene, at room temperature (Figure 1). This reaction has been previously described using the CrO_3_–pyridine complex in stoichiometric amounts [25].

Afterward, the rearrangement of dienone **12** was studied using the same conditions described in our previous report [3]. After 13 h of reaction in the presence of BF_3_.OEt_2_ in dichloromethane at room temperature, phenol ester **13** was obtained with good yield. A mechanism of simultaneous decarboxylation (see Appendix A) can explain the formation of phenol **13**, where the Lewis acid, at first, coordinates with the oxygen atom of the carbonyl group, thus causing the activation of C-9. In turn, this causes the migration of the methyl group from C-10 to C-9 with the simultaneous loss of CO_2_ and methanol, finally giving rise to phenol ester **13**. On the other hand, treatment of **12** with the I_2_/PPh_3_ system gave rise to bicyclic phenol diester **14** with a yield of 95%. The mechanism of formation of **14** from **12** probably involves the participation of iodic acid generated “in situ” by the PPh_3_–I_2_ system and water present in the reaction medium (see Appendix A).

The next step in the proposed approach for the synthesis of simplified cassane analogs was the reduction of the ester group of phenol **13**. This reaction was accomplished with LiAlH_4_ in THF at room temperature. Once hydroxyphenol **15** was obtained, the insertion of carbon monoxide through palladium-catalyzed carbonylation was carried out, providing lactone **16**, the simplified analog of pterolobirin H (**3**), with a good yield (78%) [26] (Figure 2).

In summary, the synthesis of **16**, simplified analogous of pterolobirin H (**3**), and other intermediates have been performed, starting from β-cyclocitral (**8**). All these compounds are summarized in Appendix A.

### 2.2. Anti-inflammatory Activity

#### 2.2.1. Cell Viability Assay on RAW 264.7 Cell Line

The cytotoxic effects of the simplified analogs (**10**–**16**) were analyzed on RAW 264.7 murine monocytes/macrophages by the well-established MTT (3-(4, 5-dimethyl thiazol-2-yl)-2, 5-diphenyltetrazolium bromide) assay [27] to establish the values of the sub-cytotoxic concentrations of these compounds (Table 1). Cells were treated with gradually increased concentrations of the compounds (0 to 100 μg/mL), and the viability was determined after 72 h of treatment by the absorption of formazan dye uptake, expressed as a percentage of untreated control cells. MTT is transformed into formazan only in cells with metabolic capacity (viable cells), being its absorbance proportional to the number of viable cells.

Among the tested compounds, hydroxyphenol **15** displayed the most potent cytotoxic effect, with the lowest IC_50_ value, 3.98 ± 0.94 μg/mL. The remaining compounds had weak to moderate cytotoxic activities, as indicated by IC_50_ values ranging from 35.24 ± 0.94 g/mL to 90.14 ± 1.26 g/mL (Table 1).

From these results, we determined the sub-cytotoxic concentrations corresponding to ¾ IC_50_, ½ IC_50_, and ¼ IC_50_, which were used in the next assays to ensure that the anti-inflammatory effect of our compounds was exclusively due to their anti-inflammatory properties and not to their intrinsic cytotoxicity. Next, we determined whether these compounds were able to inhibit the production of NO.

#### 2.2.2. Inhibition of NO Production Assay

Nitric oxide (NO) is released during the inflammatory response as a second messenger. NO is produced by cytokine-driven inducible nitric-oxide synthase (iNOS) as a pro-inflammatory mediator playing a key role in the immune system and is only produced when the immune cell is induced or stimulated, typically by pro-inflammatory cytokines and/or bacterial lipopolysaccharide (LPS) [28]. Therefore, NO production can be used as the main indicator to assess inflammatory activity. RAW 264.7 murine macrophages release the highest concentration of NO during their inflammatory response, being a model particularly indicated in screening studies of anti-inflammatory compounds [29,30]. These macrophages were activated with LPS for 24 h after the compounds were added. As the nitrite (NO^2–^) quantification is proportional to the NO release, the anti-inflammatory potential of the synthesized products **10**–**16** was analyzed through the Griess reaction [31] by measuring the nitrite concentrations in the cell-culture medium at sub-cytotoxic concentrations (¾ IC_50_, ½ IC_50_, and ¼ IC_50_) after 72 h of incubation. The inhibition percentages were calculated concerning the increase between the positive control (only LPS-treated control cells) and negative control (untreated control cells). The results showed that all compounds were able to strongly inhibit the NO release in a dose-dependent manner, generally with percentages of inhibition between 47.6% and 100%. The simplified analogs **10**–**13** reached a higher effect with 100% of NO inhibition at the ¾ IC_50_ concentration, followed by compounds **14**–**16** with strong NO inhibition rates: 91.8%, 86.7%, and 54.5%, respectively, at the same sub-cytotoxic concentration. At the ½ IC_50_ and ¼ IC_50_ concentrations, the inhibition of NO production, exerted by all the tested compounds, remained strong to moderate (69.5% to 47.6%) (Figure 3).

For the complete anti-inflammatory characterization of the compounds, we calculated the concentration that reduces the production of NO to 50% (IC_50 NO_) at 72 h of cell incubation. In addition, the ratio concerning Diclofenac (DCF) was calculated. DCF is a well-known NSAID (non-steroidal anti-inflammatory drug) whose IC_50 NO_ data (IC_50 NO_ = 47.12 ± 4.85 μg/mL) was previously determined by our group [32] (Table 2).

Our data showed that all compounds were more effective than DCF, with IC_50 NO_ values between 0.62 μg /mL and 27.60 μg/mL. Hydroxyphenol **15** displayed the strongest NO inhibition with IC_50 NO_ = 0.62 ± 0.21 μg/mL; this value was 76 times less than that found for DCF (IC_50 NO_ = 47.12 ± 4.815 μg/mL). The simplified analog of pterolobirin H **16** also showed strong NO inhibition with IC_50 NO_ 7.49 ± 0.15 μg/mL and was 6.29 times more potent than DCF— this was followed by compounds **12** and **14,** with IC_50 NO_ 19.44 ± 0.12 μg/mL and 11.18 ± 0.09 μg/mL, which were less than the ones found for DCF by 2.42 and 1.71 folds, respectively. Finally, compounds **10** and **11** showed an almost similar effect with IC_50 NO_ values around 20 μg/mL, remaining two times more active than DCF (Figure 4).

Inspired by our previous work focusing on the anti-inflammatory effect of a set of synthetic cassane diterpenes [16], we have established a structure–activity relationship study between these diterpenoids and their simplified analogs (see Appendix A).

In this study, we have demonstrated that the simplified analog **16** (IC_50 NO_ = 7.49 ± 0.15 μg/mL) exhibited greater inhibition of NO release than its correspondent natural pterolobirin H (**3**), (IC_50 NO_ = 17.57 ± 0.31 μg/mL), with an increase of 2.34 folds. Additionally, hydroxyphenol **15** (IC_50 NO_ = 0.62 ± 0.21 μg/mL) also showed 44.47 times more NO inhibition than its correspondent cassane relative (IC_50 NO_ = 27.57 ± 0.20 μg/mL) [16] (Appendix A).

#### 2.2.3. RAW 264.7 Cell-Cycle Assay by Flow Cytometry

The inflammatory processes induced by LPS in RAW 264.7 cells include monocyte/macrophage cell differentiation and cell-cycle arrest [13]. We analyzed the DNA ploidy and the distribution of these cells in each cell-cycle phase to investigate the effect of cassane-simplified analogs **15** and **16** (compounds with the lowest IC_50_ NO values) on the cell-cycle profile changes induced by LPS in RAW264.7 cells. For this purpose, flow cytometry by propidium iodide (PI) staining was used [33]. LPS triggers significant growth arrest of the RAW 264.7 cell cycle, producing 100% of cells detention in the G0/G1 phase (positive control: cells treated only with LPS), which is 50% higher than the negative control (untreated cells, where only 49.6% of cells were in the G0/G1 phase).

Cell-cycle analyses showed that compounds **15** and **16** could rescue LPS-induced cell-cycle arrest and promote cell proliferation in the RAW264.7 macrophages in a dose-dependent manner. Treatment with dienone **16** showed that the percentage of cells found in the G0/G1 phase was around 35% at the assayed concentrations. This decrease was accompanied by a consequent increase of S-phase cell proportions, around 60%, at the assayed concentrations. Compound **15** induced similar effects, increasing the number of cells in the G0/G1 phase to 30% at the ¼ IC_50_ and ½ IC_50_ concentrations and to 25% at ¾ IC_50_ and decreasing the S-phase cell proportion to 60%, 69%, and 75% at these concentrations. These changes were accompanied by adjustments in the G2/M phase, decreasing the number of cells in this cell-cycle phase in a dose-dependent manner. This recovery of the cell cycle concerning positive control cells could be a consequence of the anti-inflammatory effect produced by these cassane-simplified analogs, showing an important immunomodulatory potential (Figure 5).

Due to their intriguing chemical structures and biological activities, cassane diterpenoids have attracted significant attention from researchers. Their study has led to the development of new therapeutic agents or provided insights for the development of synthetic analogs with improved properties. Previous studies showed that several new cassane-type diterpenoids isolated from the seeds of *Caesalpinia minax* Hance exhibit an anti-inflammatory effect by inhibiting the expression of COX-2, except for spirocaesalmin, whose C and D rings were connected through a spiro-atom. However, bonducellpin D, a furanoditerpenoid lactone with a hydroxyl group at C-1, exhibits an important inhibition of the COX-2 enzyme compared to its analog with an additional hydroxyl group at C-2, caesalmin A, with IC_50_ values of 2.4 ± 0.1 and 3.2 ± 0.2 μM, respectively [34]. Other research found that voucapane WA, 6-acetoxyvouacapane, and vouacapenan, obtained from the seed kernels of *Caesalpinia cucullata* Roxb, exhibit an anti-inflammatory effect, inhibiting NO production in LPS-induced RAW 264.7 cells and targeting the key residues in the iNOS active cavity to diminish iNOS enzymatic activation [15].

### 2.3. Anticancer Activity

#### 2.3.1. Cell Viability Assay

Studies were conducted to evaluate the antiproliferative effects of the seven cassane-simplified analogs **10**−**16** against three cancer cell lines (B16−F10, murine melanoma; HT29, human Caucasian colon adenocarcinoma; and Hep G2, human Caucasian hepatocyte carcinoma) at increasing concentrations (0–100 μg/mL). The determination of cell viability was performed by MTT assay, and cell viability was determined after treatment for 72 h by the absorption of formazan dye. These data were expressed as a percentage concerning untreated control cells. The concentrations required for 50% growth inhibition (IC_50_) were determined for each compound (Table 3).

All the tested compounds displayed dose-dependent cytotoxic effects after 72 h of treatment. The lowest IC_50_ values were shown for hydroxyphenol **15** (2.45 μg/mL in HT29 cells, 3.25 μg/mL in HepG2 cells, and 3.84 μg/mL in B16-F10 cells). All the rest of the simplified cassanes, **10**–**14** and **16**, displayed moderate cytotoxic activities in the three cancer cell lines assayed, with IC_50_ data between 19.20 μg/mL and 93.24 μg/mL. Our results showed that pterolobirin H (**3**) displayed a weaker cytotoxic effect than its simplified analog **16**, with IC_50_ values of 70.07 μg/mL in HT29 cells, 45.31 μg/mL in HepG2 cells, and 39.71 μg/mL in B16-F10 cells [14]; showing an increase of 3.7 folds, 1.67 folds, and 1.47 folds in the IC_50_ values, respectively. Additionally, simplified analog **15** was 20-fold, 4-fold, and 13-fold more antiproliferative than its correspondent cassane diterpene **15cas** (previously synthesized and evaluated for its biological activities by our group) against the HT29, HepG2, and B16-F10 cells, respectively [16] (Figure 6).

Considering the results of cytotoxicity obtained for the simplified analogs in the three cell lines assayed, we selected hydroxyphenol **15** since it presented the lowest IC_50_ concentration value, 2.45 ± 0.29 μg/mL, in the HT29 cell line. This compound and cell line were selected for the following cytometric and microscopic studies (apoptosis characterization, changes in mitochondrial membrane potential, and morphological apoptotic changes).

#### 2.3.2. Apoptotic Assay by Flow Cytometry

The loss of cytoplasmic membrane asymmetry occurs during apoptosis, causing the translocation of phosphatidylserine (PS) from the leaflet of the internal membrane to the external membrane, exposing PS to the external environment, where it is recognized by macrophage cells. Thus, we explored whether the cytotoxic and cytostatic effects of hydroxyphenol **15** on HT29 cells were related to the induction of apoptosis through double–staining with Annexin V (An-V), conjugated fluorescein isothiocyanate (FITC), and propidium iodide (PI). The apoptosis percentages were determined at 72 h after treatment with hydroxyphenol **15** at its corresponding IC_50_ (2.45 ± 0.29 μg/mL) and IC_80_ (3.35 ± 0.30 μg/mL) previously determined concentrations for Annexin V-FICT/PI–stained cells using flow-activated cell–sorter (FACS) cytometry analysis. As illustrated in Figure 7, this double–staining method differentiated four cell populations: normal cells (annexin V-and PI–), early apoptotic cells (annexin V+ and PI), late apoptotic cells (annexin V+ and PI+), and necrotic cells (annexin V− and PI+) [35,36]. Treatment of HT29 human colon adenocarcinoma cells with analog **15** was shown to induce apoptosis in a time– and concentration–dependent manner. Compound **15** produced significant apoptotic effects on the HT29 cells, with high total apoptosis percentages reaching 40.45% (24.0% early apoptosis, 16.45% late apoptosis) at the IC_50_ concentration and 50.70% (26.25% early apoptosis, 24.45% late apoptosis) at the IC_80_ concentration. Additionally, at the used concentrations, the percentages of the necrotic population were unremarkable (Figure 7).

#### 2.3.3. Mitochondrial Membrane Potential (MMP) Assay by Flow Cytometry

The mitochondrial membrane potential (Δψm) is an important parameter of mitochondrial function and is used as an early apoptotic marker in cells [37,38]. We used it to establish the possible mechanism involved in the apoptotic responses in the HT29 cell line because of hydroxyphenol **15**-induced treatment. The apoptotic effects of anticancer agents can occur through the activation of intrinsic or extrinsic apoptotic pathways. When the mitochondria undergo disruption and are involved in the loss of the MMP, the activation is then intrinsic (mitochondrial). However, apoptosis induction without MMP alterations suggests the activation of the extrinsic apoptotic pathway [39]. After incubation with compound **15** for 72 h at the IC_50_ and IC_80_ concentrations, the changes in (Δψm) were measured using flow cytometry staining with Rh123/IP [40]. Figure 7 shows that treatment with **15** did not alter the MMP; therefore, it was unable to induce mitochondrial membrane potential depolarization in the HT29 cells, which suggests the activation of the extrinsic apoptotic pathway since, at this time and concentrations, this product was apoptotic.

Additionally, flow cytometry was used to measure DNA ploidy to determine the cytostatic effects caused by the cytotoxic response of hydroxyphenol **15** in HT29 cells (Appendix A). The distribution of cells in different cell-cycle phases was analyzed after 72 h of treatment by the incorporation of propidium iodide (PI), enabling the determination of the cell percentages in each phase of the cycle and the visualization of the cell subpopulations with different DNA contents [41]. DNA histogram analysis showed that the simplified analog **15** did not produce cell-cycle changes in the HT29 cell line concerning the control. These data are consistent with the MMP and apoptosis analysis. Therefore, we propose that hydroxyphenol **15** triggers apoptosis in HT29 colon adenocarcinoma cells by rapidly activating the extrinsic pathway without enough time to produce changes in the cell-cycle profile or the mitochondrial membrane potential (Figure 8).

#### 2.3.4. Morphological Apoptotic Changes by Fluorescence Microscopy (Hoechst Staining)

The Hoechst procedure is a fluorescent staining method that can be used to identify apoptotic cells. It involves using a fluorescent dye called Hoechst that binds to the nicked DNA in cells, a characteristic that cells exhibit in apoptotic cell death, allowing them to be visualized under a fluorescent microscope. Apoptotic cells can be distinguished by changes in their nuclear morphology, such as condensation and fragmentation, which are visible with Hoechst staining [42].

The morphological analysis of Hoechst–stained cells in HT29 cells treated with product **15** showed that they had undergone remarkable morphological changes (Figure 8). At the IC_50_ concentration of hydroxyphenol **15**, several cells showed typical apoptotic changes, including chromatin condensation, cell shrinkage, and loss of normal nuclear architecture. Furthermore, at the IC_80_ concentration, the disruption of cell–membrane integrity was more prominent. Phase–contrast light microscopy observations of fluorescence after Hoechst staining showed that a significant number of cells treated with **15** acquired apoptotic features, as evident by nuclear fragmentation and the prominent disruption of cell-membrane integrity (Figure 9).

In summary, our results showed the clear activation of apoptosis in the cytotoxic response to hydroxyphenol **15**, with characteristic morphological changes showing apoptotic cell death. The loss of mitochondrial potential at the cytotoxic and apoptotic concentrations can help provide insight into the possible molecular mechanisms involved in the apoptotic response. Thus, hydroxyphenol **15** probably activates the extrinsic apoptotic pathway. Therefore, we can propose that this apoptotic process could be triggered by the death receptor (TNF, FasL) ligand through the recruitment of the adapter protein (TRADD/FADD) that mediates the activation of caspase-8, activating, in turn, caspase-3, leading to apoptosis (Figure 10) [43,44,45,46]. Further molecular studies will be necessary to confirm these assertions.

Some current studies, with respect to apoptosis induction, have demonstrated a high correlation between the antitumor properties of cassane diterpenoids and diverse signaling pathways that trigger cellular apoptosis [47,48,49,50,51,52,53,54]. For example, phanginin D, extracted from *Caesalpinia sappan*, promoted the activity of caspase-3, along with the cleavage of procaspase-3 and poly ADP-ribose polymerase (PARP), as well as phanginin R, which induced the A2780 cells’ apoptosis in a dose-dependent manner [50,52].

Additionally, Z.Y. Li et al. demonstrated the antineoplasmic activities of caesalpin G, isolated from the CHCl_3_ extract of the seeds of *Caesalpinia minax*, against AtT-20 pituitary adenomas cells (the IC_50_ values were 26.63 ± 4.75 for 24 h and 12.77 ± 2.29 μM for 48 h) against AtT-20 cells. This cassane diterpenoid, caesalpin G, induced the AtT-20 cell apoptosis by promoting ER stress and suppressing the Wnt/β-catenin signaling pathway [54]. J. Deguchi et al. proved that a sucutinirane-diterpene derivative induces apoptosis via oxidative stress in HL-60 cells; this cassane butenolide elevated caspase 3/7 activity and decreased expression of Bcl-2 family proteins, Mcl-1, and Bid, generating reactive oxygen species in HL-60 cells [50]. A recent study reported that 3β-acetyl-nor erythrophlamide, a monocassaine diterpenoid amine from Erythrophleum fordii, triggers apoptosis via the extrinsic pathway in many cancer cell lines by inhibiting the expression of Bcl-2 and activating caspase-8 [54].

### 2.4. Molecular Docking Studies

To predict the structure–activity relationship and understand the mechanism of action for anti-inflammatory and anticancer activities, in silico molecular docking studies were investigated. Based on the IC_50_ values, the most potent cassane analogs **3**, **15**, and **16** were docked into the active site of inducible nitric-oxide synthase (iNOS) (PDB ID: 4UX6) protein and **15** into caspase 8 (PDB ID: 3KJN). The cognate ligands on the crystal structures 4UX6 and 3KJN were redocked to check the validity of the docking protocol. The favored poses of the cognate ligands matched those of the crystal structures (see Appendix A), which confirms that the chosen protocol is acceptable for reproducing the native poses. The molecular docking results for compounds **3**, **15** and **16** are presented in Table 4 and Figure 11. The docking studies showed ligands suitably located at the active sites of iNOS and caspase 8, with acceptable estimated binding energies varying from −8.15 to −7.07 Kcal/mol. A protein–ligand interaction study for product **15** reveals the presence of two hydrogen bonds with GLU371 and one with TYR367 through the hydroxyl (OH) groups. Moreover, compound **15** makes additional hydrophobic interactions with HEM, VAL346, and PRO344. These results follow the literature, which reports that GLU371 and TYR367 are among the pivotal amino acids responsible for inhibiting iNOS [55,56,57]. For compound **16**, the protection of the two hydroxyl groups has a negative effect in which hydrogen bonds were not formed between the ligand and the protein. Only hydrophobic interactions were observed with PRO344, VAL346, TYR367, and HEM. Similarly, for compound **3**, the absence of the two hydroxyl groups has a negative effect on the interactions (no hydrogen bonds were observed), which follows the experimental results. On the other hand, stabilization of analog **15** within the active site of caspase 8 occurred through four hydrogen bonds with amino acids ARG260, SER316 (x2), and GLN358. Moreover, the phenyl moiety of compound **15** provides a Pi–cation interaction with ARG413. Moreover, three hydrophobic interactions were observed between alkyl groups and CYS360, ARG413, and HIS317. Comparing our results with the literature, we noticed that the amino acids ARG260, GLN358, SER316, CYS360, and ARG413 play a crucial role in activating caspase 8 [58,59,60]. The structure–activity relationship study revealed that the two hydroxyl groups should not be modified because they make donor and acceptor hydrogen bonds. In addition, introducing aromatic groups may improve the activity by forming extra electrostatic interactions. Overall, these results suggest that the anti-inflammatory and anticancer activities of compounds **3**, **15** and **16** could be mediated by targeting iNOS and caspase proteins.

## 3. Materials and Methods

### 3.1. Chemistry

#### 3.1.1. General Information

Unless stated otherwise, the reactions were performed in oven-dried glassware under an argon atmosphere using dry solvents. The solvents were dried as follows: benzene over Na and benzophenone; dichloromethane (DCM) over CaH_2_. Thin-layer chromatography (TLC) was performed using F254 precoated plates (0.25 mm) and visualized by UV-fluorescence quenching and phosphomolybdic acid solution staining. The flash chromatography was performed on silica gel (230–400 mesh). Chromatography separations were carried out by conventional column on silica gel 60 (230–400 mesh), using hexane–ethyl acetate (AcOEt/hexane) mixtures of increasing polarity. ^1^H and ^13^C NMR spectra were recorded at 500 and 400 MHz and at 150, 125, and 100 MHz, respectively. The chemical shifts (δ H) are quoted in parts per million (ppm), referenced to the appropriate residual solvent peak and tetramethylsilane. Data for the ^1^H NMR spectra are reported as follows: chemical shift (δ ppm) (multiplicity, coupling constant (Hz), integration), with the abbreviations s, br s, d, br d. dd, hept, and m denoting singlet, broad singlet, doublet, broad doublet, triplet, double doublet, heptuplet, and multiplet, respectively. *J* = the coupling constant in Hertz (Hz). Data for the ^13^C NMR spectra are reported in terms of the chemical shift relative to Me_4_Si (δ 0.0), and the signals were assigned utilizing DEPT experiments and based on heteronuclear correlations. Infrared spectra (IR) were recorded as thin films or as solids on an FTIR spectrophotometer with samples between sodium chloride plates and were reported in the frequency of absorption (cm^−1^). Only selected absorbances (ν_max_) are reported. The ([α]_D_^25^) measurements were carried out in a polarimeter using a 1 dm-length cell and CHCl_3_ as a solvent. Concentrations are expressed in mg/mL. HRMSs were recorded on a spectrometer, utilizing a Q-TOF analyzer and ESI^+^ ionization.

#### 3.1.2. Synthesis

1,3,3-trimethyl-2-vinylcyclohex–1–ene (**9**).

To a suspension of methyltriphenylphosphonium bromide (5.46 g, 15.29 mmol) in THF (30 mL) cooled at 0 °C, 8 mL of 2 M hexane solution of *n*–butyllithium was carefully added, and the resulting mixture was stirred at 0 °C for 15 min. Next, a solution of β–cyclocitral (**8**) (1.8 g, 11.84 mmol) in THF (10 mL) was added, and the reaction mixture was stirred for 30 min. At this time, TLC showed no remaining starting material. The reaction was quenched by an addition of 5 mL of water, and the solvent was removed under vacuum. The mixture was extracted with dichloromethane (30 mL) and washed with water and brine. The organic phase was dried over anhydrous Na_2_SO_4_, filtered, and evaporated under reduced pressure; the crude product was purified by flash chromatography (hexanes) to provide diene **9** (1.6 g, 91%) as a colorless oil. The spectroscopic data of diene **9** agreed with the literature data [23].

Treatment of diene **9** with DMAD in toluene at reflux: Obtention of dimethyl 5,5–dimethyl-5,6,7,8–tetrahydronaphthalene–1,2–dicarboxylate (**10**) and dimethyl 5,5,8a–trimethyl–3,5,6,7,8,8a–hexahydronaphthalene–1,2–dicarboxylate (**11**).

Dimethyl acetylenedicarboxylate (500 mg, 3.52 mmol) was added to a solution of diene **9** (300 mg, 2 mmol) in toluene (5 mL), and the mixture was heated at reflux for 24 h. After this time, the TLC shows the complete disappearance of the starting material. The reaction was concentrated under vacuum to obtain a crude product that was purified by silica gel column chromatography using 5% AcOEt/hexane to obtain 105 mg of **10** (19%) and 331 mg of **11** (57%) as a syrup mixture. The spectroscopic data of compound **11** agreed with the literature data [23].

Treatment of diene (**9**) with DMAD in xylene at reflux: Obtention of dimethyl 5,5–dimethyl–5,6,7,8–tetrahydronaphthalene–1,2–dicarboxylate (**10**).

DMAD (240 mg, 1.69 mmol) was added to a solution of diene **9** (120 mg, 0.8 mmol) in xylene (2 mL), and the mixture was heated at 140 °C for 8 h under an argon atmosphere. At this time, TLC showed no remaining starting material. The reaction mixture was purified directly by silica gel column chromatography (3% AcOEt/hexanes) to obtain 201 mg of compound **10** (91%) as a colorless syrup. ^1^H NMR (CDCl_3_, 500 MHz) δ (ppm): 7.80 (d, *J* = 8.4 Hz, 1H), 7.45 (d, *J* = 8.4 Hz, 1H), 3.96 (s, 3H), 3.88 (s, 3H), 2.74 (t, *J* = 6.4 Hz, 2H), 1.87–1.77 (m, 2H), 1.70–1.65 (m, 2H), 1.31 (s, 6H). ^13^C NMR (CDCl_3_, 125 MHz) δ (ppm): 170.30 (C=O), 166.17 (C=O), 151.87 (C), 135.63 (C), 133.36 (C), 127.69 (CH), 127.25 (CH), 124.31 (C), 52.45 (CH_3_), 52.31 (CH_3_), 38.21 (CH_2_), 34.57 (C), 31.61 (2 xCH_3_), 27.22 (CH_2_), 18.96 (CH_2_). IR (film): 1105, 1155, 1195, 1026, 1248, 1273, 1292, 1434, 1721, 2950 cm^−1^.

Treatment of diene (9) with DMAD in the absence of solvent: Obtention of dimethyl 5,5,8a–trimethyl–3,5,6,7,8,8a–hexahydronaphthalene–1,2–dicarboxylate (**11**).

DMAD (1.3 g, 9.1 mmol) was added to diene **9** (450 mg, 3 mmol), and the mixture was sealed in a tube under an argon atmosphere and heated at 120 °C for 12 h. TLC shows the complete disappearance of the starting material. The reaction was allowed to cool to room temperature, and the mixture was purified by silica gel column chromatography using 10% AcOEt/hexane, and 665 mg (76%) of the adduct **11** was obtained as a white syrup.

Dimethyl 5,5,8a–trimethyl–3-oxo–3,5,6,7,8,8a–hexahydronaphthalene–1,2–dicarboxylate (**12**).

To a solution of **11** (760 mg, 2.6 mmol) in benzene (10 mL) was added *tert*–butyl hydrogen peroxide in decane 2.5 M (1.25 mL, 3.12 mmol) and 100 mg PDC, and the reaction mixture was stirred for 6 h at room temperature. At this time, TLC shows no starting material. The reaction was then quenched with 5 mL of sat.aq Na_2_S_2_O_3_ (5% solution) and diluted with AcOEt–H_2_O (15:5 mL). The phases were shaken and separated. Then, the organic phase was washed with water (5 mL) and brine (5 mL) and dried over anh. Na_2_SO_4_, and concentrated to obtain a crude product purification by silica gel flash chromatography using 15% AcOEt/hexane, providing dienone **12** as a white solid. (646 mg, 85%). ^1^H NMR (CDCl_3_, 500 MHz) δ (ppm): 1.21 (s, 3H), 1.31 (s, 3H), 1.38 (td, *J* = 13.2, 12.5, 4.4 Hz, 1H), 1.55 (td, *J* = 13.8, 13.1, 4.3 Hz, 1H), 1.69 (s, 3H), 1.62–1.67 (m, 2H), 1.98–1.84 (m, 2H), 3.81 (d, *J* = 11.7 Hz, 6H), 6.35 (s, 1H). ^13^C NMR (CDCl_3_, 125 MHz) δ (ppm): 17.89 (CH_2_), 27.19 (CH_3_), 29.02 (CH_3_), 32.35 (CH_3_), 34.24 (CH_3_), 37.72 (C), 39.97 (CH_2_), 42.51 (C), 52.78 (CH_3_), 52.80 (CH_3_), 124.41 (CH), 131.99 (C), 157.91 (C), 165.47 (C), 166.52 (C), 173.55 (C), 182.67 (C). IR (film): 753, 789, 880, 919, 1026, 1068, 1098, 1148, 1201, 1250, 1368, 1388, 1434, 1657, 1702, 1734, 2950 cm^−1^. HRMS (ESI) *m*/*z*: calcd for C_17_H_23_O_5_ (M + H^+^) 307.1545, found: 307.1550.

Methyl 3-hydroxy–1,5,5–trimethyl–5,6,7,8–tetrahydronaphthalene–2–carboxylate (**13**).

BF_3_·OEt_2_ (2 mL, 16.19 mmol) was added dropwise to a stirred solution of dienone **12** (319 mg, 1.04 mmol) in dry dichloromethane (5 mL) at 0 °C under an argon atmosphere. The resulting mixture was stirred for 15 h at room temperature, at which time TLC showed no starting material. The mixture was then washed with AcOEt–H_2_O (30:10 mL); the phases were separated, and the organic layer was washed with water (2 × 10 mL) and brine (10 mL) and dried over anh. Na_2_SO_4_ and filtered. Evaporation of the solvent under vacuum provides a crude product, which was purified by silica gel flash chromatography, where 5% AcOEt/Hexane was used to obtain phenol **13** (287 mg, 90%) as a yellow oil. ^1^H NMR (CDCl_3_, 500 MHz) δ (ppm): 1.29 (s, 6H), 1.60–1.65 (m, 2H), 1.78–1.88 (m, 2H), 2.40 (s, 3H), 2.59 (t, *J*. 2H), 3.96 (s, 3H), 6.90 (s, 1H). ^13^C NMR (CDCl_3_, 125 MHz) δ (ppm): 18.3 (CH_3_), 19.6 (CH_2_), 28.0 (CH_2_), 31.6 (2 CH_3_), 34.6 (C), 38.2 (CH_2_), 52.0 (CH_2_), 111.7 (C), 112.9 (C), 127.2 (C), 138.7 (C), 153.3 (C), 158.6 (C), 171.8 (C). IR (film): 723, 741, 801, 867, 949, 1042, 1077, 1138, 1192, 1203, 1221, 1256, 1289, 1332, 1436, 1564, 1660, 1730, 2929 cm^−1^. HRMS (ESI) *m*/*z*: calcd for C_15_H_21_O_3_ (M + H^+^) 249.1491, found: 249.1490.

Treatment of **12** with PPh_3_/I_2_: Obtention of (Dimethyl 3–hydroxy–5,5–dimethyl-5,6,7,8–tetrahydronaphthalene–1,2–dicarboxylate (**14**).

To a solution of PPh_3_ (96 mg, 0.36 mmol) in dichloromethane (5 mL) was added I_2_ (91 mg, 0.36 mmol), and the mixture was stirred at room temperature for 5 min. After that, the mixture was added over a solution of dienone **12** (110 mg, 0.36 mmol) in dichloromethane (5 mL). Then, the resulting mixture was stirred at room temperature for 15 h, at which time TLC showed no remaining starting material. The reaction was quenched with 10% aqueous NaHSO_3_ (1 mL), and the solvent was removed under vacuum. Then, AcOEt–water (20: 5 mL) was added, the phases were shaken and separated, and the organic phase was washed with water (5 mL) and brine (5 mL). The organic phase was dried over anh. Na_2_SO_4_ was filtered and concentrated to obtain a crude product, which was purified by silica gel flash chromatography using 5% AcOEt/hexane to obtain compound **14** (101 mg, 96%) as a colorless syrup. ^1^H NMR (CDCl_3_, 500 MHz) δ (ppm): 1.29 (s, 6H), 1.64 (m, 2H), 1.80 (m, 2H), 2.58 (m, 2H), 3.90 (s, 3H), 3.91 (s, 3H), 7.05 (s, 1H), 10.60 (s, 1H). ^13^C NMR (CDCl_3_, 125 MHz) δ (ppm):19.1 (CH_3_), 26.6 (CH_2_), 31.5 (2 CH_3_), 34.8 (C), 38.1 (CH_2_), 52.2 (CH_3_), 52.7 (CH_3_), 107.01 (C), 116.8 (CH), 124.5 (C), 134.8 (C),153.3 (C), 158.9 (C), 169.1(C), 169.8 (C). IR (film):751, 806, 1037, 1130, 1216, 1262, 1325, 1341, 1439, 1675, 1737, 2951, 3450 cm^−1^. HRMS (ESI) *m*/*z*: calcd for C_16_H_20_O_5_ Na (M + Na^+^) 315.1208, found 315.1196.

3–(hydroxymethyl)–4,8,8–trimethyl–5,6,7,8–tetrahydronaphthalen–2–ol (**15**).

To a solution of phenol **13** (830 mg, 3.34 mmol) in THF (10 mL) was added LiAlH_4_ (250 mg, 6.58 mmol), and the reaction mixture was stirred at room temperature for 8 h. The reaction was quenched with 2 mL of sat. aq. NH_4_Cl (10%) at 0 °C and the solvent was removed under vacuum. The crude reaction was diluted, and AcOEt–H_2_O (30: 10 mL) and 2N HCl (3 mL) were added. The phases were shaken and separated. The organic phase was washed with water (2 × 10 mL), brine (10 mL) and dried over anh. Na_2_SO_4_ and filtered. The solvent was evaporated under reduced pressure to obtain hydroxyphenol **15** (624 mg, 85%) as a colorless syrup. ^1^H NMR (CDCl_3_, 500 MHz) δ (ppm): 1.27 (s, 6H), 1.6–1.63 (m, 2H), 1.82–1.8 (m, 2H), 2.13 (s, 3H), 2.57 (m, 2H), 4.80 (s, 2H), 6.74 (br s, 1H), 6.77 (s, 1H). ^13^C NMR (CDCl_3_, 125 MHz) δ (ppm): 15.02 (CH_3_), 19.68 (CH_2_), 27.84 (CH_2_), 31.83 (2 x CH_3_), 34.1 (CH_2_), 38.5 (CH_2_), 66.77 (CH_2_), 111.9 (C), 118.41 (C), 126.88 (C), 135.45 (C), 147.42 (C), 153.56 (C). IR (film): 755, 864, 1048, 1164, 1220, 1273, 1312, 1361, 1382, 1421, 1460, 1575, 1603, 2864, 2924, 3349 cm^−1^. HRMS (ESI) *m*/*z*: calcd for C_14_H_19_O_2_ (M-H^+^)—219.1358, found 219.1379.

4,8,8–trimethyl–5,6,7,8–tetrahydronaphtho[2,3-b]furan–2(3H)–one (**16**).

Pd(PPh_3_)_4_ (65 mg, 0.056 mmol) and P(*o*–tolyl)_3_ (85 mg, 0.24 mmol) were introduced into a tube, and a flow of argon was passed into the tube for 5 min. Toluene (1 mL), a solution of **15** (350 mg, 1.59 mmol) in toluene (2 mL), anhydrous acetic acid (440 mg, 4.31 mmol), and formic acid (210 mg, 4.56 mmol) were then added. The resulting mixture was sealed and stirred at 100 °C for 16 h, at which time TLC showed no remaining starting material. The solvent was evaporated under vacuum affording a crude product, which was purified by flash chromatography on silica gel (15% AcOEt/hexane) to obtain lactone **16** as a white solid (285 mg, 78%). ^1^H NMR (CDCl_3_, 500 MHz) δ (ppm): 1.31 (s, 6H), 1.65–1.68 (m, 2H), 1.83–1.89 (m, 2H), 2.16 (s, 3H), 2.60–2.63 (m, 2H), 3.62 (s, 2H), 6.99 (s, 1H). ^13^C NMR (CDCl_3_, 125 MHz) δ (ppm): 16.51 (CH_3_), 31.98 (2 CH_3_), 106.10 (CH), 19.30(CH_2_), 27.38 (CH_2_), 32.53 (CH_2_), 38.47 (CH_2_), 34.61 (C), 34.61 (C), 65.88 (C), 119.73 (C), 130.30 (C), 132.76 (C), 174.71 (C). IR (film): 773, 879, 937, 1012, 1127, 1151, 1165, 1213, 1361, 1380, 1420, 1469, 1621, 1728, 1815, 2855, 2938 cm^−1^. HRMS (ESI) *m*/*z*: calcd for C_15_H_19_O_2_ (M + H^+^) 231.1385, found: 231.1374.

### 3.2. Biological Experiment

#### 3.2.1. Test Compounds

In 5 mg/mL DMSO, the products **10**–**16** were dissolved. A stock solution was conserved at −20 °C, and before the treatment, this solution was diluted in a cell-culture medium to the appropriate concentrations for each test.

#### 3.2.2. Cell Cytotoxicity Test

The mouse melanoma cells B16–F10 (ATCC no. CRL–6475), the human colorectal adenocarcinoma cell line HT29 (ECACC no. 9172201; ATCC no. HTB–38), human hepatocarcinoma cell line Hep G2 (ECACC no. 85011430), and the murine monocyte/macrophage-like RAW 264.7 cell line (ATCC no, TIB–71) were all bought from the cell bank of the University of Granada in Spain. The cells were grown in Dulbecco’s Modified Eagle’s Medium (DMEM), which also contained 2 mM glutamine, 10% heat–inactivated FCS, 10,000 units/mL of penicillin, 10 mg/mL of streptomycin (for all cancer cell lines), and 50 g/mL of gentamicin (only for RAW 264.7 cell line). The cells were incubated at 37 °C with 95% humidity and 5% CO_2_ in the air. Subconfluent monolayer cells were used in all assays.

The MTT method (Sigma, St. Louis, MO, USA) was used to evaluate the effect of the compounds on cell viability. By measuring the absorbance of MTT–dye staining on cells with a metabolism capacity, the compounds’ cytotoxicities were determined. The cells were grown to a volume of 100 L in 96-well plates at concentrations of 6.0 × 10^3^ cells/mL for the HT29 and RAW 264.7 cell lines, 5.0 × 10^3^ cells/mL for the B16–F10 cell line, and 15.0 ×10^3^ cells/mL for the Hep G2 cell line, respectively, and incubated with various products (0–100 g/mL). Finally, after 72 h, 100 μL of the MTT solution (0.5 mg/mL) in 50% of PBS and 50% of the medium was added to each well. After 1.5 h of incubation, formazan was resuspended in 100 μL of DMSO, and every concentration was evaluated in triplicate.

Then, it was assessed by absorbance at 570 nm on an ELISA plate reader (Tecan Dawn MR20-301, TECAN, Grödig, Austria) for relative cell viability. For several cytometry assays, like that of apoptosis, the cell cycle, and mitochondrial membrane potential determination, product **15** with a low IC_50_ value was selected.

#### 3.2.3. Evaluation of Nitric Oxide Concentration

In 24–well cell–culture plates, RAW 264.7 cells were supplemented with 10 g/mL of LPS and plated at 6 × 10^4^ cells per well. Products (**10**–**16**) were incubated with cells for 24 h at concentrations of ¾ IC_50_, ½ IC_50_, and ¼ IC_50_, respectively. The Griess reaction was performed on a 96–well plate; 150 µL of the test samples from the supernatant or a sodium nitrite standard of 0–120 µM were mixed with 25 µL of Griess reagent A (0.1% n-(1-naphthyl) ethylenediamine dihydrochloride) and 25 µL of Griess reagent B (1% sulfanilamide in 5% phosphoric acid). The absorbance was measured at 540 nm using an ELISA plate reader (Tecan Sunrise MR20–301, TECAN, Grödig, Austria) after 15 min of incubation at room temperature. The nitrite concentration in the supernatant of each experimental sample was determined by comparing the absorbance to the standard nitrite curve. The increase in NO production between the negative control (untreated cells) and positive control (cells only treated with 10 g/mL of LPS) was determined to be 100%.

#### 3.2.4. Cell–Cycle Test (RAW264.7)

The amount of DNA present in cells can be determined effectively using PI staining flow cytometry. Flow cytometry determines the changes in cell–cycle profiles and the characteristic changes in DNA levels that occur during cell–cycle arrest and differentiation. Fluorescence–associated cell sorting (FACS) at 488 nm on an Epics XL flow cytometer (Coulter Corporation, Hialeah, FL, USA) was used to count the number of cells at each stage of the cell cycle. For this test, 12 × 10^4^ RAW264.7 murine macrophage/monocyte cells stimulated with LPS were plated on 24–well plates containing 1.5 mL of medium and incubated with the products **15** and **16** for 24 h at ¾ IC_50_ and ½ IC_50_ concentrations, respectively. The treatments were as follows: Positive control: cells treated with LPS only; negative control: only cells (untreated with anything); sample: cells treated with LPS and then treated with the tested compounds **15** and **16**. After being washed twice with PBS, the cells were harvested through trypsinization, resuspended in TBS 1X (10 Mm Tris, 150 Mm NaCl), and Vindelov buffer (100 mM Tris, 100 Mm NaCl, 10 mg/mL RNAse, and 1 mg/mL PI) was added at a pH of 8.0. The samples were placed on ice for 15 min. The cells were stained with 20 μL of a PI solution containing 1 mg/mL right before the FACS analysis. The multicycle software (multicycle AV OCX version Phoenix Flow Systems. San Diego, CA, USA) was used to analyze the data and determine how many cells were in each phase of the cell cycle—G0/G1, S, and G2/M. Means and standard deviations of at least two experiments conducted in triplicate for each concentration were used to represent the percentage of cells.

#### 3.2.5. Cell–Cycle Test (HT29)

The procedure was carried out by flow cytometry after propidium iodide staining (PI). HT29 cells were plated in 24–well plates at a density of 5 × 10^4^ cells/well, with 1.5 mL of medium, and allowed to grow for 24 h before being treated with compound **15** for 72 h at the IC_50_ and IC_80_ concentrations. Then, the cells were washed twice with PBS, trypsinized and resuspended in 1 × TBS (10 mM Tris and 150 mM NaCl), and subsequently, Vindelov buffer (100 mM Tris, 100 mM NaCl, 10 mg/mL Rnase, and 1 mg/mL PI, at pH 8) was added. Next, the cells were stored on ice and were stained with 20 µL of 1 mg/mL PI solution just before measurement. In each test, about 10 × 10^3^ cells were analyzed. The tests were carried out three times, with two replicates per test. In the end, the samples were examined using a flow cytometer, and the number of cells in each stage of the cell cycle was evaluated by fluorescence–associated cell sorting (FACS) at 488 nm in an Epics XL flow cytometer (Coulter Corporation, Hialeah, FL, USA).

#### 3.2.6. Apoptosis Test (HT29)

To confirm that compound **15** had a pro–apoptotic effect, flow cytometry was used to detect the double-staining of annexin V and PI. Using a Coulter Corporation (Hialeah, FL, USA) FACS (fluorescence–activated cell sorter) flow cytometer, apoptosis was measured by flow cytometry. For this test, 5 × 10^4^ HT29 cells were plated on 24–well plates with 1.5 mL of medium and incubated for 24 h. After that, the cells were treated with the selected compound in triplicate for 72 h at the concentrations IC_50_ and IC_80_. Then, the cells were collected and resuspended in a binding buffer (10 mM HEPES/NaOH, pH 7.4, 140 mM NaCl, 2.5 mM CaCl_2_). At room temperature in darkness, Annexin V–FITC conjugate (1 μg/mL) was then added and incubated for 15 min. The cells were stained with 5 μL of the 1 mg/mL PI solution just before the analysis. In each test, roughly 10 × 10^3^ cells were evaluated in duplicate.

#### 3.2.7. Mitochondrial Membrane Potential Test (HT29)

Analytical flow cytometry with dihydrorhodamine (DHR) was used to examine the electrochemical gradient across the mitochondrial membrane (DHR). For this test, 5 × 10^4^ HT29 cells were plated in a 24-well for 24 h, and for 72 h, they were treated with compound **15** at their corresponding IC_50_ and IC_80_ concentrations. Then, a fresh medium containing DHR at a concentration of 5 mg/mL was added. After being incubated for 1 h at 37 °C with 5% CO_2_ and 95% humidity, the cells were washed and resuspended in PBS containing 5 µg/mL of PI. The FACScan flow cytometer was used to evaluate the fluorescence intensity (fluorescence-activated cell sorter). Two replicates were used in each of the three tests.

#### 3.2.8. Hoechst–Stained Fluorescence Microscopy Test (HT29)

Using Hoechst–stained fluorescent microscopy, morphological alterations were examined. Thus, in 24–well plates, 5 × 10^4^ HT29 cells were plated on coverslips. The cells were cultured for 72 h at their respective IC_50_ and IC_80_ values after 24 h of product **15** in addition. After that, the cells were washed two times with PBS, and for 3 min, they were treated in cold MeOH, then washed with PBS, and subsequently, for 15 min in the dark, incubated in the 500 µL Hoechst solution (50 ng/mL) in PBS. Using a DAPI filter and fluorescence microscopy (DMRB, Leica Microsystems, Wetzlar, Germany), the samples were seen.

#### 3.2.9. Statistical Studies

Nonlinear regression was used to fit the experimental cytotoxicity data to a sigmoidal function (y = y_max_/(x/a)^−b^). Interpolation was used to calculate the IC20, IC50, and IC80 value concentrations that caused 80%, 50%, and 20% of the cell cytotoxicity, respectively. The SigmaPlot^®^ 12.5 software was used to carry out these analyses. Comparable examinations were performed to acquire the IC_50_ of NO exhibition (IC_50 NO_). The data presented here are all representative of at least two separate experiments that were carried out in triplicate. All quantitative information was summed up as the means ± standard deviation (SD).

#### 3.2.10. Experimental Section of Docking Studies

Furthermore, in silico computational docking studies were performed using AutoDock 4.2. The X-ray crystallographic structure of iNOS and caspase 8 was downloaded from the RCSB Protein Data Bank (RCSB PDB) ID: 4UX6 and 3KJN, respectively. The proteins were prepared separately by removing water and co–crystalized ligands bound with the protein to make the receptors free of any ligands before docking. Then, polar hydrogen and Gasteiger charges were added using the MGL tools (Molecular Graphics Laboratory), and the proteins were saved in the PDBQT format (Protein Data Bank, Partial Charge (Q), and Atom Type (T)). The structures of the ligands were created separately using ChemDraw Ultra 12.0, the energy was minimized in Chem3D, and the torsional bonds of ligands were set flexibly and saved in the PDBQT format. Next, the receptor was kept rigid, and the grid covering all the amino acid residues present inside the active sites of proteins was built (for iNOS: grid box size of 40 Å × 40 Å × 40 Å with a spacing of 0.375 Å between the grid points and centered at 21.702 (x), 33.04 (y) and 42.191 (z)). The best conformers were searched by the Lamarckian genetic algorithm (LGA), the population size was set to 150, and the maximum number of energy evaluations was set to 25,000,000. Finally, the results were analyzed and visualized by discovery studio.

## 4. Conclusions

The present work has reported the first function–oriented synthesis (FOS) of simplified analog **16**, of pterolobirin H (**3**), from β–cyclocitral (**8**) in only six steps, inspired by our previous strategy for the synthesis of this natural cassane diterpenoid. This concise and versatile process has avoided the use of protecting groups and has laid a foundation for further SAR studies of the cassane skeleton. The pharmacological evaluation showed that all the tested compounds were able to inhibit NO production more than diclofenac. This drastic and rational simplification of the hydrophilic core structure of the cassane skeleton was shown to produce potential NO inhibition. The most important NO inhibitors were analog **16**, which was twice as active as pterolobirin H (**3**), and hydroxyphenol **15**, as the most active compound, with 45 times higher activity than its corresponding synthetic cassane diterpene **15cas**, with the lowest IC_50 NO_ = 0.62 ± 0.21 μg/mL. As a result of the anticancer evaluation, we found that simplified lactone **16** exhibited the highest cytotoxic effects than pterolobirin H (**3**) in the three cancer-assayed cell lines. Moreover, the flow cytometry assays demonstrated that hydroxyphenol **15** (IC_50_ = 2.45 ± 0.29 μg/mL in HT29 cells) displayed strong apoptotic effects through an extrinsic pathway, reaching 50.70% at the IC_80_ concentration. Compound **15** showed a promising affinity to the active site of iNOS and caspase 8 via the formation of hydrogen bonds and hydrophobic interactions. Meanwhile, compounds **3** and **16** showed only hydrophobic interactions with the iNOS protein. Further in vitro and in vivo analyses of the detailed mechanism for the biological activities of cassane diterpenoids and their simplified analogs are currently under investigation in our laboratories, and efforts to synthesize more potent analogs inspired by hydroxyphenol **15** bioactivities will be described in due course.

## Data Availability

Data are contained within the article or Appendix A.

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
