# Peer review of "Synthesis of Tricyclic Pterolobirin H Analogue: Evaluation of Anticancer and Anti-Inflammatory Activities and Molecular Docking Investigations"

_molecules, 2023, doi:10.3390/molecules28176208_

Round 1

Reviewer 1 Report

In the current study, the authors focused on biological evaluation of a tricyclic pterolobirin H analogue, including molecular docking investigations. The authors have thoroughly described the chemical synthesis of Tricyclic Pterolobirin H analogue with appropriate supporting evidence.  This work is considered a novelty and appropriate for publication.

Hers are some significant comments that authors need to address before accepting.

(1).

Intermediate 15 is found to be more active than compared to target pterolobirin H analogue. What is the author's main object to the synthesis of compound 16?

(2).

I have not seen the biological activity data of pterolobirin H (3) in the current experiments and the authors claimed that the synthesized compound 16 is more active than compared pterolobirin H (3). It would be good if the authors include the activity data of pterolobirin H.

(3).

Introduction: Authors should discuss how the terpenoid natural products and their new chemical entities will play significant role in drug discovery. Please refer to the following articles. European Journal of Medicinal Chemistry, 114, 2016, 293-307 (https://doi.org/10.1016/j.ejmech.2016.03.013) & Mol Divers   19, 2015, 745–757 (https://doi.org/10.1007/s11030-015-9621-3).

(4).

How stable compound 16 analogue comparatively pterolobirin H (3)?

(5).

What is the importance of methyl group in the phenyl ring in pterolobirin H or compound 16?

(6).

Supporting information: Please provide the NMR solvent and instrument Hz to run the spectra. For example., 1HNMR (500 MHz in CDCl3).

NA

Author Response

REVIEWER #1:

(1)   Intermediate 15 is found to be more active than compared to target pterolobirin H analogue. What is the author's main object to the synthesis of compound 16?

Response:

the main objective for the synthesis of compound 16 is to study its bioactivity and compare it with that of natural pterolobirin H (3).

 (2) I have not seen the biological activity data of pterolobirin H (3) in the current experiments and the authors claimed that the synthesized compound 16 is more active than compared pterolobirin H (3). It would be good if the authors include the activity data of pterolobirin H.

Response:

In accordance with the reviewer's suggestion, the activity data of pterolobirin H (3) is included in the section of results and discussion

(3).

Introduction: Authors should discuss how the terpenoid natural products and their new chemical entities will play significant role in drug discovery. Please refer to the following articles. European Journal of Medicinal Chemistry, 114, 2016, 293-307 (https://doi.org/10.1016/j.ejmech.2016.03.013) & Mol Divers   19, 2015, 745–757 (https://doi.org/10.1007/s11030-015-9621-3).

Response:

According to the reviewer, this point has been changed in the manuscript.

“It is well known that natural products constitute the main axis for the rational design of new drugs [1,2]. However, the scarcity of these in their natural sources and their structural complexity is in many cases an important limitation to evaluate their biological potentials. The preparation of simple analogues of natural products is one of the important strategies used to study the structure-activity relationship (SAR) and also to develop new drugs more economically and with lesser difficulty. Diterpenes with cassane skeleton are an example of natural products whose biological activities have been poorly studied largely due to their difficult chemical synthesis [3]. However, these metabolites continue to attract the attention of researchers due to the broad spectrum of their pharmacological activities including antitumor, anti-inflammatory, antimicrobial, antiplasmodial, antimalarial, antiviral, antioxidant, antipyretic, antiperiodic, anthelmintic, antinociceptive, and antidiabetic properties [4]”

(4) How stable compound 16 analogue comparatively pterolobirin H (3)?

Response:

The pterolobirin H analog 16 shows high stability, similar to that of natural product.

(5) what is the importance of methyl group in the phenyl ring in pterolobirin H or compound 16?.

Response:

In our previous work (Zentar, H.; Jannus, F.; Medina-O’Donnell, M.; Lupi, A.; Justicia, J.; Alvarez-Manzaneda, R.; Reyes-Zurita, F. J.; Alvarez-Manzaneda, E.; Chahboun, R. Synthesis and Biological Evaluation of Cassane Diterpene (5α)-Vuacapane-8(14), 9(11)-Diene and of Some Related Compounds. Molecules. 2022, 27, 5705) SAR data analysis showed that the methyl group at the C-14 positions of cassane diterpenoids is not always important for their cytotoxic and anti-inflammatory activities. On the other hand, the methyl, vinyl, or carboxymethyl functional group linked at C-14 position has been described as a common structural characteristic in most of the anti-inflammatory cassanetype diterpenoids (Dong, R.; Yuan, J.; Wu, S.; Huang, J.; Xu, X.; Wu, Z.; Gao, H. Anti-Inflammation Furanoditerpenoids from Caesalpinia Minax Hance. Phytochemistry 2015, 117, 325–331). This fact is consistent with the results presented in our latest work about NO-release inhibition after 72 h, since the synthesized cassane diterpenoid pterolobirin H with a methyl group at C-14, exhibited more NO inhibition than its 14-desmethyl analogs 25. (Zentar, H.; Jannus, F.; Medina-O’Donnell, M.; Lupi, A.; Justicia, J.; Alvarez-Manzaneda, R.; Reyes-Zurita, F. J.; Alva-rez-Manzaneda, E.; Chahboun, R. Synthesis and Biological Evaluation of Cassane Diterpene (5α)-Vuacapane-8(14), 9(11)-Diene and of Some Related Compounds. Molecules. 2022, 27, 5705).

Regarding the antiproliferative activity, It is observed that the presence of the C-14 methyl group is also important for the cytotoxic effect of the pterolobirin H, against HepG2 and B16F10 cell lines; however, it seems to be unimportant for the cytotoxic effect against HT29 cells. This conjecture agrees with the results of  Pitsinos et al. The identification of Gli-Mediated Transcription Inhibitors through Synthesis and Evaluation of Taepeenin D Analogues. Medchemcomm 2016, 7, 2328–2331. B). Chatzopoulou, M.; Antoniou, A.; Pitsinos, E.N.; Bantzi, M.; Koulocheri, S.D.; Haroutounian, S.A.; Giannis, A. A Fast Entry to Furanoditerpenoid-Based Hedgehog Signaling Inhibitors: Identifying Essential Structural Features. Org. Lett. 2014, 16, 3344–3347).

(6).

Supporting information: Please provide the NMR solvent and instrument Hz to run the spectra. For example., 1HNMR (500 MHz in CDCl3).

Response:

According to the reviewer, this point has been included in the supporting information.

Reviewer 2 Report

This manuscript describes the preparation and biological properties of compounds derived from Pterolobirin H. The results obtained are clearly and logically presented and represent an interesting extension of the current state of the art. Nevertheless, I recommend to eliminate the following shortcomings:

The structures of compound 3 in Figure 1 and Scheme 1 are not identical, please correct.

The preparation of compound 10 has already been described in the literature (JOC, 1973, 38, 399, triethylene glycol dimethyl ether, 200 °C, 14 h, 100% yield) This work should be cited and briefly discussed in the context of the present observation.

The same applies to compound 12, which is again known and was also prepared by oxidation. I would recommend to cite this work and comment on the differences in the synthesis.

The biological activity of compound 3 is briefly mentioned in the introduction. I would list the biological activities of compound 3 for the tables that describe the biological activity of compounds 1016.

Figure 3 and 5 have a strange quality (looks like a scanned or edited image). I recommend inserting a picture with better resolution.

The "Molecular docking studies" chapter is missing the study for compound 3 as it is the parent compound.

All compound numbers in the experimental section must be bolded.

It looks like the NMR spectra were processed in MNova, which inserts the coupling constants (J) as normal text. But J must be in italics.

line 533, there is "tertbutyl hydrogen peroxide" should be "tert-butyl hydroperoxide"

line 563, there is "PPh3/I2" should be "PPh3/I2"

Line 581: The name of the compound 15 is strange.

Line 595: The name of compound 16 is strange.

A dash not a hyphen should be used for separation of multiplet boundaries in description of 1H NMR spectra

I recommend the above changes in ESI:

1H NMR spectra should be listed with a range of 10-0 ppm

13C NMR spectra should be listed with a range of 200-0 ppm.

The mechanism for the formation of compound 14 is incorrect. The intermediate has protonated carbonyl group. However, where does the proton come from? Does the proton come from DCM or compound 12? In my opinion, I don't see any other proton source in the chemical equation.

Compound 13 has a different quality of 13C NMR spectrum compared to the 1H spectrum. In addition, the purity judged by the 1H and 13 spectra is different. Was a different sample used for the measurement? Please provide the quality of the spectrum.

 Also, compound 15 has a 13C NMR spectrum of below average quality. Please supply a good quality spectrum.

Author Response

REVIEWER #2:

(1)       The structures of compound 3 in Figure 1 and Scheme 1 are not identical, please correct.

Response:

According to the reviewer, the structure of compound 3 was corrected.

(2)       The preparation of compound 10 has already been described in the literature (JOC, 1973, 38, 399, triethylene glycol dimethyl ether, 200 °C, 14 h, 100% yield) This work should be cited and briefly discussed in the context of the present observation.

Response:

According to the reviewer, the preparation of compound 10 and reference (JOC, 1973, 38, 399) has been included (Page 4, lines 131-134).

(3) The same applies to compound 12, which is again known and was also prepared by oxidation. I would recommend to cite this work and comment on the differences in the synthesis.

Response:

According to the reviewer, the oxidation of compound 11  and reference (S.P. Tanis, K. Nakanishi, J. Am. Chem. Soc. 1979, 101, 15, 4398–4400, https://doi.org/10.1021/ja00509a071 has been included (Page 4, lines 136-138).

(4) The biological activity of compound 3 is briefly mentioned in the introduction. I would list the biological activities of compound 3 for the tables that describe the biological activity of compounds 1016.

Response:

The description of the biological activity of pterolobirina H (3) has not been detailed in the introduction for the simple reason that there are not many works that describe said activity. The biological activities of pterolobirin H (3) have been included in the section of results and discussion.

(5) Figure 3 and 5 have a strange quality (looks like a scanned or edited image). I recommend inserting a picture with better resolution.

Response:

According to the reviewer, figures 3 and 5 have been modified and the low quality (resolution) has been improved.

(6) The "Molecular docking studies" chapter is missing the study for compound 3 as it is the parent compound.

Response:

The molecular docking study for compound 3 has been included, in the Results and experimental sections

(7) All compound numbers in the experimental section must be bolded.

Response:

According to the reviewer, all compound numbers in the experimental section have been bolded.

(8) It looks like the NMR spectra were processed in MNova, which inserts the coupling constants (J) as normal text. But J must be in italics.

Response:

According to the reviewer, all coupling constants (J) are italicized.

(9) line 533, there is "tertbutyl hydrogen peroxide" should be "tert-butyl hydroperoxide"

Response:

According to the reviewer, TBHP name has been corrected.

(10) line 563, there is "PPh3/I2" should be "PPh3/I2"

Corrected

(11) Line 581: The name of the compound 15 is strange.

Corrected

(12) Line 595: The name of compound 16 is strange.

Corrected

(13) A dash not a hyphen should be used for separation of multiplet boundaries in description of 1H NMR spectra

Corrected

(14) 1H NMR spectra should be listed with a range of 10-0 ppm

Response:

According to the reviewer, 1H NMR spectra are listed with a range of 10-0 ppm.

(15) 13C NMR spectra should be listed with a range of 200-0 ppm.

Response:

According to the reviewer, 13C NMR spectra are listed with a range of 200-0 ppm.

(16) The mechanism for the formation of compound 14 is incorrect. The intermediate has protonated carbonyl group. However, where does the proton come from? Does the proton come from DCM or compound 12? In my opinion, I don't see any other proton source in the chemical equation.

Response:

I agree with the reviewer, it seems that the mechanism is correct, but it is not stated where the proton comes from. in my opinion the proton comes from iodic acid which is generated by the I2/PPh3 system and water. In the reaction we do not add water, but it seems that using DCM without drying and operating in open air is sufficient to generate an equivalent of iodidric acid in the medium. To make sense of the proposed mechanism, H2O has been added in the scheme of the mechanism and also mentioned in the manuscript (page 5, lines 150-152).

(17) Compound 13 has a different quality of 13C NMR spectrum compared to the 1H spectrum. In addition, the purity judged by the 1H and 13 spectra is different. Was a different sample used for the measurement? Please provide the quality of the spectrum. Also, compound 15 has a 13C NMR spectrum of below average quality. Please supply a good quality spectrum.

Response:

I agree with the reviewer. Better quality spectra have been provided.

Reviewer 3 Report

A good work was exerted here by the authors. However, some important points need to be addressed. So, a major revision may be required for the paper improvement:

1-      A systematic abstract should be provided. For example, the abstract should start with general background about pterolobirin H. In addition, the abstract should briefly speak about a problem raised by the authors and how this work could solve it. So, please make the abstract more interesting for readers.

2-      The title should be informative declaring the main therapeutic effects of the synthesized compounds as anti-inflammatory and anti-cancer candidates.

3-      The green highlights in the keywords sections, need to be removed.

4-      Scheme 1, could be replaced to be Figure 2 (rather than Scheme 1), since it is just an illustration.

5-      Please define DMAD abbreviation for the first time only.

6-      Scheme 2 and 3, please put the yield % under each compound (Not below the arrows).

7-      The resolution of most figures needs improvement.

8-      Please define DCF abbreviation for the first time only.

9-      The number formatting in Table 3 need to be corrected.

10-   Please define MMP abbreviation for the first time only.

11-   It is highly recommended to compare in vitro anticancer properties with a reference anticancer drug.

12-   Why molecular docking was conducted on Caspase 8?

13-   Where are the IR and mass spectrometry charts in the supplementary material???

14-   Some old references need to be updated.

15-   Regarding 1H NMR charts, the total peak integrations indicating the total number of protons are not compatible with the total number of protons of the expected chemical structure for some compounds (e.g Compound 15 and 16). That could be attributed to issues in compound purity. Please check

16-   Usually, we do not need to cite references within the conclusion part.

Author Response

REVIEWER #3:

  • (1) A systematic abstract should be provided. For example, the abstract should start with general background about pterolobirin H. In addition, the abstract should briefly speak about a problem raised by the authors and how this work could solve it. So, please make the abstract more interesting for readers.

Response:  A systematic abstract has been provided.

  • (2) The title should be informative declaring the main therapeutic effects of the synthesized compounds as anti-inflammatory and anti-cancer candidates.

Response:  The title has been changed according to the reviewer recommendation.

  • (3) The green highlights in the keywords sections, need to be removed.

Response:  According to the reviewer, the highlights in the keywords sections have been removed.

  • 4- Scheme 1, could be replaced to be Figure 2 (rather than Scheme 1), since it is just an illustration.

Response:  According to the reviewer, Scheme 1, has been replaced to be Figure 2.

(5) Please define DMAD abbreviation for the first timeonly. Corrected

  • (6) Scheme 2 and 3, please put the yield % under each compound (Not below the arrows).

Response:  According to the reviewer the % have been put under each product

  • (7) The resolution of most figures needs improvement.

Response:  According to the reviewer, The resolution of all figures has been improved.

  • (8) Please define DCF abbreviation for the first time only.

Response:  DFC abbreviation has been defined only for the first time

  • (9) The number formatting in Table 3 need to be corrected.

Response: We agree with the reviewer and the number formation in the table 3 has been corrected.

  • (10) Please define MMP abbreviation for the first time only.

Response:  In agreement with the reviewer MMP abbreviation has been defined only for the first time.

  • (11) It is highly recommended to compare in vitro anticancer properties with a reference anticancer drug.

Response:  In this type of assay, we used only negative control (untreated cells) corresponding to 100% of cell viability to compare the compounds effects. We think that to use a product that reduce 100 % of cell viability does not provide any information. However, we agree with the reviewer and these indications will be considered in future studies, but unfortunately, now, we do not have the requested results.

  • (12) Why molecular docking was conducted on Caspase 8?

Response: Molecular docking was conducted on caspase 8, since the proposed apoptotic molecular mechanism could be triggered by the death receptor (TNF, FasL) ligand through the recruitment of the adapter protein (TRADD/FADD) that mediates the activation of caspase-8, activating in turn caspase-3, leading to apoptosis. However, thanks to the reviewer question we reconsidered the molecular docking conducted on caspase 8 deleted it.

  • (13) Where are the IR and mass spectrometry charts in the supplementary material???

Response: Added the IR data for compounds

  • (14) Some old references need to be updated.

Response: We thank the reviewer for his valuable observation; the old references have been cited in this work because they represent the only works that exist in the literature to discuss similarities, antecedents and strategies related to the main objectives of this study.

  • 15- Regarding H NMR charts, the total peak integrations indicating the total number of protons are not compatible with the total number of protons of the expected chemical structure for some compounds (e.g Compound 15 and 16).That could be attributed to issues in compound purity. Please check

Response: We have checked the 1H NMR data for these compounds. the total peak integrations are a little different than the total number of protons of compounds 15 and 16, probably due to the presence of small impurities.

  • (16) Usually, we do not need to cite references within the conclusion part.

Response:

We agree with the reviewer, the reference cited within the conclusion part has been eliminated.

Round 2

Reviewer 2 Report

Well done.

Author Response

We thank the reviewer 

Reviewer 3 Report

I thank the authors for correcting most points raised. However, the authors should keep the molecular docking part conducted on caspase 8 illustrating the reason of caspase  8 choice (authors declared in their response) to make the established in silico studies more comprehensive. 

Author Response

According to the reviewer's indication, the molecular docking part performed on caspase 8 has been preserved. Both in the manuscript and in the supporting information, the molecular docking part conducted on caspase 8 is highlighted in green.